# The Influence of Copper on Halogenation/Dehalogenation Reactions of Aromatic Compounds and Its Role in the Destruction of Polyhalogenated Aromatic Contaminants

**Tomáš Weidlich**

Chemical Technology Group, Institute of Environmental and Chemical Engineering, Faculty of Chemical Technology, University of Pardubice, Studentska 573, 53210 Pardubice, Czech Republic; tomas.weidlich@upce.cz; Tel.: +420-46-603-8049

**Abstract:** The effect of copper and its compounds on halogenation and dehalogenation of aromatic compounds will be discussed in the proposed article. Cu oxidized to appropriate halides is an effective halogenation catalyst not only for the synthesis of halogenated benzenes or their derivatives as desired organic fine chemicals, but is also an effective catalyst for the undesirable formation of thermodynamically stable and very toxic polychlorinated and polybrominated aromatic compounds such as polychlorinated biphenyls, dibenzo-*p*-dioxins and dibenzofurans accompanied incineration of waste contaminated with halogenated compounds or even inorganic halides. With appropriate change in reaction conditions, copper and its alloys or oxides are also able to effectively catalyze dehalogenation reactions, as will be presented in this review.

**Keywords:** halogenation; oxybromination; oxychlorination; copper-catalyzed; arylation; Ullmann reaction; coupling; hydrodehalogenation

## 1. Introduction

Aromatic carbon–halogen ($C_{arom}$–X) bonds are common functional groups in organic synthesis. They are frequently and widely applied in the preparation of numerous organic fine chemicals. In many cases, halogenated aromatic compounds (Ar-Xs) are valuable intermediates for subsequent $C_{arom}$–R bonds formation. The broad application of Ar-Xs is joined, however, with a high risk of environmental pollution by almost non-biodegradable and toxic Ar–Xs which need subsequent chemical treatment of polluted matter with the aim of converting Ar–Xs to more-biodegradable non-halogenated products. Due to this reason, even the facile cleavage of $C_{arom}$–X with the aim of producing nonhalogenated and usually more biodegradable $C_{arom}$–H or $C_{arom}$–R has been intensively studied.

In this article, the research advances on the copper-catalyzed and mediated $C_{arom}$–X (X = F, Cl, Br, I) bond formation via direct $C_{arom}$–H bond transformation and (hydro)dehalogenation of $C_{arom}$–X producing $C_{arom}$–H or $C_{arom}$–R, respectively.

Copper and its salts exhibit broad catalytic activity mainly due to the easily accessible and reasonable stability of Cu(0), Cu(I), Cu(II) and Cu(III) oxidation states. They are therefore effective catalysts not only for $C_{arom}$–H bonds transformations by single electron transfer (SET) processes [1,2] but through using appropriate conditions even for $C_{arom}$–halogen cleavage reactions by SET, two-electron transfer, or other mechanisms [3,4]. Copper also ranks among the cheap, earth-abundant first-row transition metals which are more environmentally acceptable (relatively less toxic) metals in comparison with noble platinum group metals, gold or silver. Copper-mediated reactions have also gained significant popularity in recent times. The excellent functional group tolerance of copper-mediated reactions offers new opportunities for the synthesis of a large variety of organic compounds.

This review intends to cover most of the recent advances both on Cu-based halogenation and dehalogenation reactions of aromatic compounds based on C–O and C–C bond formation.

## 2. Cu-Based Halogenations of Aromatic Compounds

This section may be divided by subheadings. It should provide a concise and precise description of the experimental results, their interpretation as well as the experimental conclusions that can be drawn. Aryl halides are widely used in organic synthesis to form carbon-carbon and carbon-heteroatom bonds under metal catalysis such as in Heck, Sonogashira, Suzuki, and Ullmann coupling reactions [5]. Aryl halides are also highly versatile synthetic intermediates for many applications in agrochemicals, pharmaceuticals and materials [6,7]. Aryl iodides are generally more reactive in organic transformations. Although aryl chlorides or aryl bromides are relatively more inert, they are much more commonly found in pharmaceuticals and agrochemicals, in which they are introduced to modify the physical and biological properties of aromatic rings [8].

Traditional methods for Ar–Xs syntheses involved two common preparatory routes: direct halogenation via an electrophilic substitution reaction (SEAr) and a nucleophilic aromatic substitution reaction (SNAr) especially of diazonium salts [9].

Halogenation of aromatic compounds using highly toxic chlorine ($Cl_2$) and bromine ($Br_2$) via electrophilic aromatic substitution (SEAr) reaction mechanism requires special caution with regards to handling and safety. In addition, corresponding toxic and very corrosive hydrogen halides (HXs) are produced as by-products during halogen-based halogenations in stoichiometric quantity. In addition, HX as an undesirable by-product (a) diminishes significantly X atom efficiency of halogenation processes based on $X_2$ as a halogenation agent (X atom economy is max. 50%) and (b) causes difficulties with subsequent HX treatment.

The development of convenient and more efficient methods using a source of halide ion combined with different oxidants for the synthesis of aryl and heteroaryl halides has therefore attracted increasing attention [10–20].

A different way for increasing the Cl atom economy in the production of chlorinated aromatic compounds via SEAr exploiting gaseous $Cl_2$ is based on the ex-situ oxidative treatment of co-produced hydrogen chloride. Electrolytic and catalytic technologies can be applied for this purpose. HCl electrolysis is based on the conversion of a 22 wt.% HCl solution over graphite electrodes separated by a diaphragm to $Cl_2$ and $H_2$, respectively at the anode and the cathode. High electricity consumption leads to the unfavorable operating costs of this process [21].

The catalytic option is based on the gas-phase oxidation of HCl to $Cl_2$ by $O_2$ over heterogeneous Cu- or other transition metal-based catalysts [22].

### 2.1. The Role of Cu-Catalyzed $O_2$-Based Oxidation of Waste HCl Producing $Cl_2$

The so-called Deacon process, Cu-based oxidation of waste HCl with $O_2$ is one of the intensively studied techniques enabling reuse of chlorine spent during chlorination processes from HCl by-product (Scheme 1) [23–25].

$$2\ HCl + O_2 \xrightarrow{\text{Cu(I)/Cu(II)}} Cl_2 + H_2O$$

**Scheme 1.** Cu-catalyzed oxidation of HCl at 360–450 °C (Deacon process) [23–25].

The Deacon process is catalyzed by $CuCl_2/CuO$ with a mechanism that is generally accepted involving three steps (Equations (1)–(3)) [24]:

$$2\ CuCl_2 \rightarrow 2\ CuCl + Cl_2 \tag{1}$$

$$2\,CuCl + O_2 \rightarrow CuO + CuCl_2 \tag{2}$$

$$CuO + HCl \rightarrow CuCl_2 + H_2O \tag{3}$$

The Cu-based recycling of chlorine from waste HCl significantly increases Cl atom efficiency of $S_EAr$ chlorination process and is an important part of chlorination technologies [25]. The other disadvantage of $S_EAr$-based synthetic strategies is the low regioselectivity of this halogenation and problems with the formation of more halogenated products (Scheme 2).

**Scheme 2.** Electrophilic aromatic substitution ($S_EAr$) pathway.

Progress for the selective synthesis of aryl and heteroaryl halides has recently attracted increased attention. Significant advances dealing with higher regioselectivity have occurred in the $C_{arom}$–H halogenation catalyzed by different transition metals such as Pd, Rh, Ru, Au [26], Ag [27–29], etc. In the area of $C_{arom}$–H bond halogenation, the copper catalysis also serves a useful utilization. As a class of cheap and simply available and widely used transition metal catalysts, Cu-based salts have exhibited substantial application in $C_{arom}$–H bond functionalization in recent years owing to their distinct advantages such as low toxicity, high stability and flexible forms of presence [30–34].

Aerobic oxybromination of arenes catalyzed by $Cu(NO_3)_2$ was achieved using HBr as a bromine source, molecular oxygen as the oxidant and water as a solvent with high selectivity. The catalyst not only demonstrates high chemoselectivity for monobromination, but also remarkable regioselectivity for para-isomers [35]. The proposed mechanism is similar to the Deacon process, bromine is produced by $O_2$-based Cu(II)-catalyzed oxidation of HBr (Scheme 3).

**Scheme 3.** Proposed mechanism of oxybromination [35].

This procedure represents an attractive green approach to brominated aromatics for several reasons: (i) it avoids the use and handling of $Br_2$; (ii) no VOCs (volatile organic compounds) are used as solvents; (iii) the components in the reaction, air and water are non-toxic, HBr is much less toxic than $Br_2$, reagents are non-flammable, cheap and readily available. Therefore, this method offers an efficient system for the organic solvent-free oxybromination of aromatics with high atom economy and selectivity. The $Cu(OAc)_2$ catalyzed technique for oxybromination of phenols substituted with electron-donating (Edg) groups using LiBr was published by Menini et al. [36]. The authors suggest a SET mechanism for this bromination reaction. Similarly, anilines substituted with both Edg and Ewg were selectively brominated using $Cu(OAc)_2$ catalyst [37] by a SET mechanism (Scheme 4). The first step consists of the formation of a Lewis acid-base complex **A** between Cu(II) and aniline or phenol. We then suggest the abstraction of a hydrogen atom and electron transfer to Cu(II) resulting in radical **B**, Cu(I) and H+. Tautomeric cyclohexadienyl radical **C**

reacts with Cu(II) bromide giving Cu(I) bromide and brominated product **D**, whose tautomerization gives para-bromoaniline or para-bromophenol **2**. Re-oxidation of Cu(I) complexes by dioxygen readily occur in acetic acid solutions and complete the catalytic cycle (Scheme 5).

$$2 \bigcirc\!\!-A\text{-H} + O_2 + 4\,H^+ + 2\,Br^- \xrightarrow[\text{80 °C in AcOH}]{Cu(OAc)_2} 2 \bigcirc\!\!-A\text{-H} + 2\,H_2O$$

A = O or NH

**Scheme 4.** Regioselective bromination of anilines and phenols [36,37].

**Scheme 5.** The suggested single electron transfer (SET) mechanism for the bromination of anilines or phenols [36,37].

Apart from soluble Cu(II)-salts, the heterogeneous Cu-based halogenation variant with Cu-phtalocyanine encapsulated on zeolite was published using hydrogen peroxide or oxygen as an oxidant [38].

Stahl and co-workers distinguished the mechanisms of the regioselective Cu-catalyzed chlorination and bromination of arenes and heteroarenes in 2009 [39]. Good to excellent yields of the corresponding mono-halogenated products were obtained by using lithium halides and $O_2$ as the halogen sources and the terminal oxidant, respectively. The brominations occurred more readily than the corresponding chlorinations and in some cases, the latter required higher amounts (even superstoichiometric) of $CuCl_2$. The regioselectivities obtained therein matched those expected of an electrophilic halogenation which is controlled by sterics. The authors also noted that the bromination reactions turned red-brown in color upon heating, in addition to the fact that cyclooctene undergoes dibromination in a trans fashion under the reaction conditions. Electrophilic aromatic substitution ($S_EAr$) was proposed for the bromination reactions proceeding via direct attack of $Br_2$ which is produced from $CuBr_2$ under the reaction conditions (Scheme 3). The mechanism of the chlorination was in contrast proposed to occur via electron transfer from the electron-rich arene to $CuCl_2$. Analogously, Yang reported the same mechanism during a para bromination of aniline using $CuBr_2$ in combination with Oxone as the oxidant [40].

## 2.2. The Role of Directing Groups in Regioselective Cu-Catalyzed Halogenation of Aromatic Compounds

Highly regioselective ortho-halogenations of substituted 2-phenylpyridines catalyzed by Cu(II) salts were described by several research groups [41–44]. Pyridine works as a monodentate directing group in this case (Scheme 6). Each of these methods is based on oxyhalogenation of accessible halide using either strong chemical oxidant ($CrO_3$ in

method A) or oxygen at elevated temperature (methods B–D. Surprisingly, 1,2-dihalogenoethane or 1,1,2,2-tetrahalogenoethane (PHE, method B) or benzoyl chloride (method C) play an important role as the source of halogenating agent, although, they are usually used as alkylating (PHE) or acylating (PhCOCl) agents. This could be explained by the mechanism given in Scheme 7, both mentioned PHE and PhCOCl are sources of halide for Cu(II) catalyzed halogenation, presumably. In addition, benzoic acid anhydride formed by hydrolysis of PhCOCl works as the precursor of catalytically active Cu-complexes **F** (Scheme 7, X = Cl) and **G** (Scheme 7, where X = Cl and OCOPh) in halogenation according to Scheme 6, method C [43].

**A**: 4 eq. LiCl + 20 % Cu(NO$_3$)$_2$/CrO$_3$/AcOH/150$^{\circ}$C [41]
**B**: Y$_2$CHCHY$_2$ + 20 % CuCl$_2$ or Cu(OAc)$_2$+O$_2$/130$^{\circ}$C [42]
**C**: PhCOCl + 20 % Cu(OAc)$_2$+O$_2$+Li$_2$CO$_3$/toluene/145$^{\circ}$C [43]
**D**: 3 eq. LiX + 20 % Cu(NO$_3$)$_2$/AcOH+O$_2$/150$^{\circ}$C [44]

**Scheme 6.** Selective ortho-halogenation of phenyl ring bound in 2-phenylpyridine [41–44].

**Scheme 7.** Proposed mechanism of ortho-halogenation of 2-phenylpyridine catalyzed by CuX$_2$ (X = bromide, chloride or anion of carboxylic acid) [41–44].

Both SET mechanism [41] and two-electron transfer with the dominant role of Cu(III) species [44] were proposed as the main mechanism for these ortho-halogenations using CuX$_2$ depending on the used reaction conditions (Scheme 7).

In addition, it must be mentioned that not only ortho-halogenation occurs using conditions described in Method B in Scheme 6. The addition of a suitable source of nucleophile enables ortho-functionalization of the phenyl ring in 2-phenylpyridine structure, especially if non-halogenated solvent (DMSO, CH$_3$CN) was used instead of polychloroethane. Different sources of nucleophiles were used, concretely I$_2$ (for ortho-iodination), CH$_3$SiCN (for ortho-cyanation), H$_2$O (for ortho-hydroxylation), organic disulfides (for corresponding ortho-thioether formation, etc.).

In all these above-mentioned C$_{arom}$–H halogenation protocols with pyridine as the monodentate directing group (MDG), the chemo-selectivity remained a challenge since either mixed mono- and dihalogenated products or only one of the two potential products could be acquired. In this regard, a synthetic approach allowing the tunable synthesis of mono- and dihalogenated products was highly desirable. Recently, Han and co-workers [45] successfully achieved this kind of tunable reaction via a CuX-mediated C$_{arom}$–H halogenation with the assistance of *N*-halosuccinimide (NXS, where X = Cl or Br). The application of different organic acids, which participated in the in-situ formation of acyl hypohalites enabled the selective generation of mono- or di-halogenated products (Scheme 8).

**Scheme 8.** Tunable ortho-halogenation of 2-phenylpyridine catalyzed by CuX in co-action of different carboxylic acids [45].

The heterogeneous Cu-MnO-catalyzed option of selective monohalogenation of aromatic compounds with NXS in the presence of molecular oxygen, under irradiation with visible light, was also reported. Monochlorination, bromination, and iodinations were achieved in a quite selective fashion [46].

Apart from the issue of selectivity, another major challenge in the MDG-assisted C–H activation lied in the removal of the MDG, which undermined the efficiency of the synthetic procedure. To alternate the hardly removable MDG of the pyridine ring, Carretero and co-workers [47] discovered highly regioselective Cu-catalyzed ortho-monochlorination and monobromination of anilines containing a removable *N*-(2-pyridyl)sulfonyl (SO$_2$Py) auxiliary. In the presence of the copper(II) halide catalyst and NXS, a class of *o*-chloro/bromoanilines was efficiently provided under aerobic conditions (Scheme 9). The SO$_2$Py group could be removed by treatment with elemental magnesium in methanol.

**Scheme 9.** Ortho-halogenation of anilines acylated by SO$_2$Py used as monodentate directing group with subsequent of SO$_2$Py using magnesium in methanol [47].

More recently, Shi and co-workers [48] reported the *ortho*-C–H halogenation of aryl-2-carboxamides using (2-(pyridine-2-yl)isopropylamine (PIP) as a bidentate directing group (BDG). The copper catalyst combined with NXS (X = Cl, Br, I) and a proper additive promoted smoothly the synthesis of various *o*-haloaryl-2-carboxamides (Scheme 10). This synthetic protocol tolerated not only carbon aryls, but also various heteroaryls such as thiophene, furan and pyridine in the amide component.

**Scheme 10.** Ortho-halogenation of aromatic or heterocyclic carboxylic acids derivatized to appropriate amide working as a bidentate directing group (BDG) using the PIP group [48].

In the optimization process, $Zn(OAc)_2$ additive reveals a beneficial effect on the reaction yield as well as on monoselectivity. Common functional groups such as F, Cl, Br, $NO_2$ and CN are well tolerated. Apart from PIP as BDG, Li with co-workers described $Cu(OAc)_2$ catalyzed selective aerobic oxidative ortho-chlorination of benzamides with trichloroacetamide using 1-(2-Aminophenyl)-1*H*-pyrazole as a less expensive, removable BDG [49]. The used BDG was removed readily with cerium ammonium nitrate oxidation (Scheme 11).

**Scheme 11.** Ortho-halogenation of benzoic amides containing 1-(2-aminophenyl)-1-*H*-pyrazole as BDG with subsequent oxidative cleavage [49].

Liu, Wan, and Du achieved a CuO-catalyzed, one-pot N-acylation with subsequent C5–H halogenation of 8-aminoquinolines by direct employment of acyl halides as both the acylation and the halogenation reagent by co-action of air [50] (Scheme 12). The 8-Aminoquinoline skeleton works as both the substrate for acylation with acyl halide and after acylation of $NH_2$-group as BDG. This remarkable reaction is a representative example of 100% atom economy.

**Scheme 12.** Acylation of 8-aminoquinoline accompanied by selective Cu-catalyzed halogenation with proposed reaction mechanism [50].

In the first step, acylation of amino group proceeds with contemporary formation of HX which subsequently reacts with CuO catalyst and corresponding Lewis acid-base complex **E** between N-acyl-8-aminoquinoline and CuX$_2$ is generated. After single electron transfer from Cu(II) cation to aromatic ring radical cation **RC** is formed and subsequent halide migration causes formation of the probably most stable radical **R**. Radical **R** decomposes to the 5-halogenated N-acyl 8-amino-quioline **P**, hydrogen radical and CuX. Hydrogen radical reduces CuX$_2$ producing hydrogen halide and CuX. CuX$_2$ is recycled by aerobic oxidation of CuX and can take part in the repeated catalytic cycle.

Chen and co-workers developed CuX-mediated selective monohalogenation of 2-methylquinolines producing corresponding heterobenzyl halides [51] (Scheme 13). It should be noted that traditional methods for halogenation at the heterobenzyl position with NCS, NBS, or NIS result in mixtures of the monohalogenated product and the dihalogenated side product.

**Scheme 13.** Cu-catalyzed $\alpha$-halogenation of 2-methylquinolines [51].

In recent years, the formation of C–F chemical bonds received global research interest due to the particular functions of many fluorinated chemicals [2]. Accordingly, C–H fluorination reactions also become an issue of broad concern as such a transformation provides a straightforward route for rapid synthesis of diversity-enriched fluorinated products.

Daugulis and co-worker [52] established a Cu(I) catalyzed $C_{arom}$–H ortho-fluorination of N-(quinolin-8-yl)benzamides (Scheme 14). The mono- and/or difluorination took place in the presence of CuI, N-methylmorpholine N-oxide (NMO) and pyridine by using DMF as the medium and AgF as the fluorine source, providing mono- or difluorinated products, respectively. This method manifests excellent functional group tolerance and provides a novel route for the preparation of ortho-fluorinated benzoic acids. As occurred in most cases involving the activation of two identical C–H bonds, the unsatisfactory chemoselectivity in forming mixed products in many entries continued to be a problem to address [52] (Scheme 14).

**Scheme 14.** CuI/AgF based ortho-fluorination of benzamides bearing quinoline-based BDG [52].

### 2.3. Alternative Methods Applicable for Cu-Catalyzed Syntheses of Ar-Xs

In 2002, Subramanian reported the conversion of aromatics to monofluoroaromatics using a CuF$_2$ catalyst [53] (Scheme 15). The CuF$_2$ oxidatively fluorinates the aromatic substrate with concomitant formation of HF and Cu metal. The latter is reoxidized to the active CuF$_2$ species by two equivalents of HF and a half equivalent of O$_2$, generating one equivalent of water as the sole by-product in the process. Despite the high temperatures and narrow scope of this process, it served as an important starting point for more advanced catalytic methods of halogenation using copper.

**Scheme 15.** High-temperature CuF$_2$ catalyzed fluorination with subsequent recycling of CuF$_2$ by oxyfluorination of produced copper [53].

Transition metal-catalyzed or transition metal-mediated halogen exchange has recently been emerging as a promising pathway for the production of otherwise not readily available Ar-Xs [54–57].

Aryl iodides (Ar–Is) have recently attracted increased attention due to their high reactivity in comparison with the corresponding Ar-Cls or Ar-Brs and the possibility of producing valuable hypervalent iodine-based reagents applicable as oxidants, etc. [58,59].

Cu-catalyzed decarboxylative iodination of 2-nitrobenzoic acids to the corresponding 2-nitro-iodobenzenes was published by Stang et al. (Scheme 16) [60].

**Scheme 16.** Cu(II)-catalyzed decarboxylative iodination of 2-nitrobenzoic acid [53].

A general method for the interconversion of different aromatic halogen derivatives (so-called aromatic Finkelstein reaction) was also described converting aryl chlorides or bromides into the corresponding aryl iodides or aryl fluorides using the corresponding Cu(I) halogenide as the catalyst and N,N′-dimethyl-cyclohexane-1,2-diamine **L1** (Figure 1) as a ligand [61] (Scheme 17).

**Scheme 17.** CuI catalyzed aromatic halogen-exchange reaction (aromatic Finkelstein reaction) [61].

**Figure 1.** Structures of ligands used for aromatic Finkelstein reaction [55,61–63].

A wide array of substrates, including sterically hindered aryl bromides and heteroaryl bromides were well tolerated under the reaction conditions which employ inexpensive CuI, excess of NaI and a commercially available racemic diamine as a supporting ligand [55]. Buchwald and co-workers reported a potentially scale-up attractive variation to the above described Cu-catalyzed aromatic Finkelstein reaction which used a continuous flow reactor [62]. Remarkably, it was observed that the reaction of PhBr performed with KI catalyzed by CuI/1,10-phenanthroline in DMF at 110oC resulted in the rapid formation of iodobenzene in good yields [63]. The subsequent kinetics studies, conducted by reacting an equimolar mixture of PhBr and KI or PhI and KBr, both afforded a mixture with an identical composition (20% PhBr and 80% PhI), thereby indicating the existence of thermodynamic equilibrium between PhI and PhBr (Scheme 18).

**L2** (10 mol%)
CuI (5 mol%)

Ph—Br + K**I** ⟶ $\xrightarrow[\text{110 °C, 48 h}]{\text{DMF}}$ Ph—**I** + KBr
80 %

**Scheme 18.** CuI catalyzed halide exchange reaction [63].

The interconversion of different aromatic iodo- or bromo derivatives into Ar-Br or Ar-Cl was also described. Utilization of ionic liquid 1-butyl-3-methylimidazolium bromide (BMIMBr) as the solvent by co-action of copper halide salts enables, for example, this halogen-exchange reaction (Scheme 19) [64].

**Scheme 19.** CuX based iodide exchange reaction in ionic liquid BMIMBr [64].

Another method bypassing the usage of high excess of expensive ionic liquid in the dual role of the reaction solvent and liquid ion-exchanger is based on $Cu_2O$-catalyzed conversion of aryl and heteroaryl bromides to the corresponding chlorides by cheap $Me_4NCl$ as the halide source in ethanol (Scheme 20) using **L3** as the ligand (Figure 1). This reaction provides good to excellent yields of the aryl and heteroaryl chloride products when 20 mol % of [Cu] is employed [65].

**Scheme 20.** $Cu_2O$ catalyzed bromide exchange reaction using quaternary ammonium chloride [65].

These interconversion methods make potentially possible in-situ preparation of reactive Ar-I derivatives from common and simply available Ar-Cl or Ar-Br derivatives for subsequent nucleophilic aromatic substitution of halogen. These reactions are described in the following chapters.

It can be stated that Cu-based halogenations of aromatic compounds enable the modern regioselective formation of a wide range of Ar-Xs which are applicable as intermediates for the synthesis of organic fine chemicals. Massive production and broad utilization of halogenated aromatic compounds are also joined with the production of hazardous waste streams often contaminated with stable polychlorinated by-products which could cause long-term contamination of the environment.

### 2.4. Cu-Mediated Formation of Polychlorinated Aromatic Pollutants

Disposal of hazardous wastes containing chlorinated aromatic compounds in a way that minimizes the environmental hazards has become an urgent issue. Conventional incineration of these wastes produces harmful compounds, such as polychlorinated biphenyls (PCBs), polychlorinated dibenzo-p-dioxins and polychlorinated dibenzofurans

(PCDD/Fs) and polychlorinated naphtalenes (PCNs); even in slight concentrations, their stability, bioaccumulation ability and toxicity are enormous. Generally, PCNs, PCBs and PCDD/Fs are always emitted from thermal-related industries and these chemicals all have similar physicochemical properties, biological effects and toxic mechanisms [66]. Their formation is connected with Cu-catalyzed $O_2$-based oxidation of HCl (Deacon reaction) and Cu-catalyzed C-O coupling [67,68] (Scheme 21).

**Scheme 21.** Cu-catalyzed de-novo synthesis of chlorinated aromatic compounds (Ar–Cl) [67–74].

It has been suggested in several studies that the so-called de novo synthesis of PCBs, PCNs and PCDD/Fs occurs in thermal-related industries from the coupling of phenoxy radicals (formed from chlorinated phenols) or from the degradation of polycyclic aromatic hydrocarbons followed by their repeated chlorination at a temperature above 300 °C and the co-action of oxygen. Chlorination can be catalyzed by transition metal species, especially copper compounds (Scheme 21), which are known to have strong catalytic effects [69–74]. These chemicals all have similar physicochemical properties, biological effects and toxic mechanisms [66]. It was observed, for example, that unexpectedly high concentrations of PCNs were formed on fly ash from a secondary copper smelting process at 250–450 °C [74].

For the above-mentioned reasons, dehalogenation reactions of Ar-Xs not only play an important role for the transformation of Ar–Xs to the desired organic fine chemicals (drugs, pesticides, etc.) but even for the transformation of harmful and non-biodegradable Ar-Xs to the dehalogenated and obviously readily biodegradable non-halogenated products. Copper-based catalysis plays an important role even in this field. Due to the focus of this article on the applicability of copper in context with the solution of the environmental aspects of application of halogenated aromatic compounds, only Cu-catalyzed C-C and C-O couplings together with reductive dehalogenation reactions are reviewed as the crucial processes for the possible chemical degradation of polyhalogenated oxygen heterocycles and their above-mentioned precursors.

### 3. Cu-Catalyzed Dehalogenation of C_arom-X Bond

*3.1. The Role of Cu in Nucleophilic Substitutions of Halogens in Ar-Xs*

Cu-catalyzed arylation reactions devoted to the formation of C–C and C–heteroatom bonds (Ullmann type couplings) have acquired great importance in the last two decades. From the environmental chemistry point of view, arylation reactions converting Ar-Xs to the corresponding non-halogenated products are also an uncoverable way for effective chemical destruction of very stable and almost non-biodegradable and toxic halogenated aromatic compounds. The scope of Cu-catalyzed cross-coupling reactions is increasing, and seems to be somewhat complementary to that of Pd-based methodologies. Finally, in many cases, Cu-catalyzed reactions work well without any ligand, and when required, the ligands are usually structurally quite simple and inexpensive (ligands for Pd chemistry are often complex, expensive and air-sensitive) [75,76].

As the ligands applicable for complexation of catalytically active Cu(I) complexes applicable for C-heteroatom bonds formation, different secondary (N,N′-dimethylethanediamine (DMEDA), N,N′-dimethylcyclohexane-1,2-diamine **L1**) or tertiary 1,2-diamines (1,10-phenanthroline **L2** or N,N,N′,N′-tetramethylethanediamine (TMEDA)), enamines, oximes, phosphines, aminoacids (proline **L3**, N,N-dimethylglycine **L7**), thiofene-2-carboxylic acid, 1,3-dicarbonyl compounds, or diols (ninhydrin, in some cases even monosaccharides) were demonstrated for the effective course of the above-mentioned C-C or C-heteroatom formation [77,78] (Figures 1 and 2). Two different roles were attributed to the used additives for Cu-catalyzed Ar-Xs substitution reactions. The formation of more soluble cuprate species and/or stabilization of crucial Cu(I) active species for subsequent coupling reaction was suggested and/or an additive would facilitate an oxidative addition of Ar-X to the Cu atom, thus accelerating the coupling [79].

The main drawback of this type of coupling reaction is the known inertness of aryl chlorides to the Cu-catalyzed nucleophilic substitution in most cases, typically only the aryl iodides and sometimes bromides are applicable for the described arylation reactions.

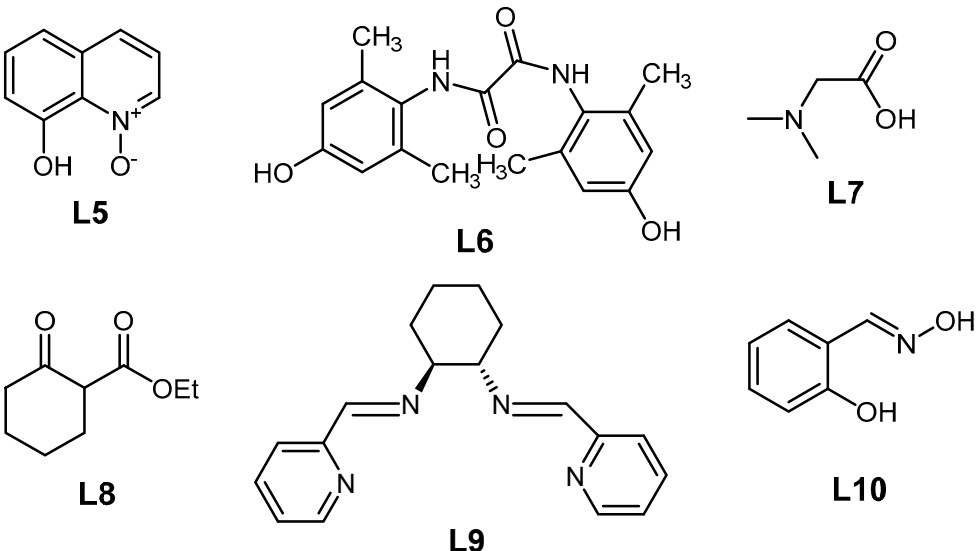

**Figure 2.** Commonly used ligands applied for Cu-catalyzed Ullmann reactions.

*3.2. Reactions of Ar-Xs with O-Nucleophiles*

3.2.1. Cu-Catalyzed Syntheses of Phenols

Phenols are important building blocks in chemical, pharmaceutical and material sciences [80]. One of the main methods used for the synthesis of functionalized phenols includes halogenation of substituted benzene derivatives followed by the nucleophilic substitution of activated aryl halides (alkali hydrolysis). Recently, several homogeneous and

heterogeneous catalysts based on copper complexes were reported for the cross-coupling of aryl halides with hydroxide salts.

The possibility of performance of Cu-catalyzed synthesis of phenols by nucleophilic substitution of Ar-Xs by OH⁻ in aqueous solutions emphasizes the applicability of this type of reaction in the green chemistry area. Application of water as an eco-friendly solvent enables not only utilization of common water-soluble bases such as sodium or potassium hydroxides or carbonates and sometimes even saccharides or glycolic acid as the suitable ligands but also utilization of water as the source of the nucleophile. For these purposes, aqueous solutions of dimethyl sulfoxide (DMSO) serve as the bio-based organic and with water fully miscible solvent [81–85].

The application of Cu(II) salt using glucose as the ligand for nucleophilic aromatic substitution of halogen in different Ar–Xs by OH⁻ nucleophile was published (Scheme 22) [83].

**Scheme 22.** Formation of phenol-based on Cu-catalyzed nucleophilic aromatic substitution using glucose as the ligand [83].

Aryl iodides and bromides 1 were reacted with excess potassium hydroxide (4–8 equivalences) in the presence of copper(II) acetate (5 mol %) and glucose (5 mol %) to give good to excellent yields of the corresponding phenols 2.

The reactivity of aryl chlorides depended on the nature of the electron-withdrawing group with substrates containing a nitro group providing an excellent yield of phenol **2** [81].

Similarly, using proline as the ligand, dihydroxyterephtalic acid is produced from corresponding sodium salt of dihalogenated terephtalic acid using $CuBr_2$ as a catalyst in an alkaline aqueous solution (Scheme 23) [82].

**Scheme 23.** Cu-catalyzed formation of 2,5-dihydroxyterephtalic acid from 2,5-dihalogenoterephtalic acid [82].

Methylated monosaccharide quebrachitol was successfully tested as the ligand of choice using $Cu_2O$ and NaOH for effective hydroxylation of aryl iodides in boiling water or aq. ethanol on air [85].

In addition, $Cu(CH_3CN)_4X$ and $Cu(NH_3)_4X$ complexes (X = $PF_6$, $HSO_4$) were proved as effective catalysts for hydroxylation of o-bromo- or o-chlorobenzoic acid in an alkaline aqueous solution. The main role of these complexes is in the increasing of a catalytically acting Cu(I) ions concentration in an aqueous reaction medium, as was documented by Saphier et al. (Scheme 24) [86].

**Scheme 24.** Cu-catalyzed formation salicylic acid from 2-halogenobenzoic [87].

Aryl bromides and even aryl chlorides were converted to the corresponding phenols in an alkaline aqueous DMSO solution using CsOH as a base under argon after several days of heating at 130°C using 8-hydroxyquinoline-N-oxide as the ligand [87].

One of the most efficient copper-based catalytic systems for hydroxylation of even aryl chlorides based comprising copper(II) acetylacetonate and N,N'-bis(4-hydroxy-2,6-dimethylphenyl)-oxalamide in DMSO/H$_2$O mixture using LiOH or KOH was described by Ma et al. [88].

Another possibility for the effective performance of different Ar–I or Ar–Br substitution reactions in the aqueous reaction medium requires phase-transfer catalysis (PTC) conditions using a two-phase (aqueous-organic) reaction mixture. In this case, CuI nanoparticles were used with co-action of Bu$_4$NOH as the source of nucleophile [89].

In conclusion, even Cu-catalyzed hydroxylation of Ar-Xs is feasible in green solvents as are DMSO/H$_2$O and cheap bases as are potassium or sodium hydroxides, carbonates or phosphates, only a few examples of effective utilization of Ar-Cls were published using entirely different ligands. In this case, the high potential of tandem catalysis to transform non-reactive Ar-Cl to reactive Ar-I that is subsequently hydroxylated to the corresponding Ar-OH could be expected.

### 3.2.2. Cu-Catalyzed Syntheses of Aryl Ethers from Ar-Xs

Cu-catalyzed nucleophilic substitution of halogen (especially I or Br) in Ar-X by aliphatic alcoholates are usually performed in polar aprotic solvents such as N,N-dimethylformamide (DMF). The mechanism of the Ar-Xs substitution by sodium methanolate as O-nucleophile was studied, CuOCH$_3$ (or NaCu(OCH$_3$)$_2$, respectively) was proposed as the key intermediate in anisole synthesis from bromobenzene by Aalten and others [90].

Methanolysis is applicable even for the preparation of sterically hindered derivatives which has been shown by Righi et al. in the case of the flavonoide Mosloflavone synthesis (Scheme 25) [91].

**Scheme 25.** Cu-catalyzed methoxylation of sterically hindered 6-bromo-5-hydroxy-7-methoxyflavone [91].

1,10-Phenanthroline **L2** was tested as a simple and cheap ligand suitable for the preparation of an effective CuI-based catalyst for the reaction of primary and secondary alcohols with iodoarenes under quite mild reaction conditions enabling the preparation of optically pure chiral ethers (Scheme 26) [92].

**Scheme 26.** CuI catalyzed coupling of chiral sec-phenethyl alcohol with retention of configuration [92].

It is known that the addition of suitable ligands to the Cu(I) salt enables increasing solubility and stabilization of catalytically active Cu(I) species and facilitates aromatic substitution by O-nucleophiles. This fact sometimes enables the utilization of common aromatic hydrocarbons (toluene, xylene) or alcohols as the solvents and lower reaction temperature for highly efficient preparation of Ar–ORs (Scheme 27).

**Scheme 27.** Cu(I)-catalyzed coupling of aliphatic alcohols with Ar-Xs (X = Br, I) [93–96].

Using 3,4,7,8-tetramethyl-1,10-phenanthroline **L4** as the ligand with $KF/Al_2O_3$ as the base (in the presence of a large excess of phenol), the coupling of alcohols with aryl iodides has also been realized [93]. Using amino acids **L3**, **L7**, **L11** (Figure 2) as ligands provides interesting results, but here the issue of having the nucleophile (the alcohol) in excess remains [94]. Especially **L4** was proved to be a highly efficient ligand, which enables this reaction to be carried out with aryl iodides under mild conditions [95,96]. Their method is particularly noteworthy for not needing a large excess of alcohol and for being remarkably selective for O-arylation when amino alcohols are involved.

In the case of preparation of diphenyl ethers, a mild heterogeneous system (reaction at 50–60 °C) based on copper nanoparticles (0.1 eq. of nano Cu (18 nm)), for the coupling of aryl iodides and aryl bromides with phenols was developed by Kidwai et al. (Scheme 28) [97].

**Scheme 28.** Copper-catalyzed coupling of phenols with Ar-Xs (X = Br, I) [97–105].

The reactions take place without any added ligand and the system is reusable, although a decrease in yield is observed and a longer time reaction is required for each subsequent run.

With the requirement of utilizing non-polar solvents, however, there are only limited reports of Cu-catalyzed O-arylation reactions being performed in toluene or xylene [98]. Mao and Wang [99] recently reported that a diamine-type ligand anchored to silica and chelated to copper, efficiently promotes the synthesis of diaryl ethers from aryl iodides, aryl bromides, or even activated aryl chlorides at 130 °C.

Amino acids were successfully tested as ligands for Cu(I) catalyzed arylation of Ar-X. Although, in principle, a number of amino acids could facilitate copper-catalyzed coupling of aryl halides and phenols, N,N-dimethylglycine **L7** was found to be the best ligand for this biaryl ether formation. Ma et al. [100] described an efficient catalytic C–O coupling reaction at ambient temperature. A particular advantage of the ambient-temperature conditions used with this system is that the coupling can occur without any accompanying racemization of the tyrosine derivatives. Using the amino acid **L7** as a supporting ligand with 30% copper loading, they could couple 2-bromotrifluoroacetanilide and L-tyrosine derivatives in high yields even at 25 °C [100]. It was observed in this reaction that an ortho-amide substituent ($CF_3CONH$-) bound on the aryl halide (2-bromotrifluoroacetanilide) is required as MDG. The additional stabilization of the CuI center, provided by O coordination of this chelating ortho group, was proposed to be a key factor for the success of these reactions.

The catalytic system based on beta-keto ester **L8** as a supporting ligand affords the corresponding diaryl ethers from aryl bromides and iodides at mild temperatures in DMSO [101]. Similarly, the imine **L9** and salicylaldoxime **L10** (Figure 2) are also very efficient at promoting the synthesis of diaryl ethers. In combination with 10% CuI and the inexpensive base $K_3PO_4$ (instead of commonly used $Cs_2CO_3$), these ligands allow for the coupling of a large range of aryl bromides with phenols under mild conditions [102,103]. The coupling of phenols with aryl bromides is also possible with **L2** as the supporting ligand. The latter was used in the presence of copper impregnated into charcoal under microwave irradiation [104]. Very effective Cu(I)-based catalysts soluble in common organic solvents were produced by complexing of $Cu(Ph_3P)_3Br$ with 1,10-phenanthroline or neocuproine and evaluated for facile cleavage of Ar-Xs with different nucleophiles [105]. Even a ligand-free copper-catalyzed diaryl ether synthesis based on arylation of Ar–I or Ar–Br with phenols using the inexpensive base $K_3PO_4$ and phase transfer catalyst $Bu_4NBr$ are typically carried out in polar, aprotic solvents (DMF) after tens of hours of action at a temperature above 120 °C [106].

The arylation in 100 kg scale of 4-bromo-chlorobenzene using 4-methoxyphenol was reported under the above-described conditions using **L7** (Figure 2) as the cheap ligand of choice producing exclusively 4-chlorophenyl-4'-methoxyphenylether. In this case, the coupling reaction worked well at 90 °C in dioxane using $Cs_2CO_3$ as a base, and a wide range of aryl iodides, aryl bromides and phenols could be used as coupling partners [107].

Aryl chlorides are in general considered too inert for C-O cross-coupling with phenols. The comparison in the reactivity of different halogens in aryl halides was studied and compared using $Cu(Ph_3P)_3X$ catalyst [108]. It was observed using the $Cu(Ph_3P)_3I$ in the reaction of 4-chloro-bromobenzene with 4-methylphenol. The aryl chloride side is at least 30-times less reactive in comparison with the aryl bromide side of 4-chloro-bromo-benzene. The obtained reaction mixture contains only 1.9% yield of 4-bromo-4'-methyl-diphenylether and below 1% of double-coupled product bis-1,4-(4-methylphenoxy)benzene) and 66.9% yield of 4-chlorophenyl-4'-methyl-phenylether.

A significant advance was achieved with the development of an efficient method for the copper-catalyzed arylation of phenols by aryl chlorides (Scheme 10) using ligand 2,2,6,6-tetramethyl-3,5-heptadione **L12** (Figure 3). This reaction system is one of the few able to condense both activated and deactivated aryl chlorides. Although relatively harsh conditions are used (135 °C), this procedure has genuine economic advantages in that very inexpensive materials are used (Scheme 29) [109].

**Figure 3.** Ligands are applicable for coupling phenols with different Ar-Xs.

**Scheme 29.** (Nano)copper catalysts-based coupling of aryl chlorides with phenols [109–112].

The other published methods enable Cu-catalyzed coupling or aryl chlorides with phenols applied CuI nanoparticles [110], CuI/sparteine **L13** [111], nano $Cu_2O$ [112] were achieved for diaryl ether synthesis starting from chlorobenzene.

In comparison with hydroxylation of aryl chlorides, above mentioned copper-catalyzed arylation of phenols with Ar-Cls seems to be more feasible due to the broader choice of effective catalytic systems, though working in non-aqueous solvents. On the other hand, in the area of treatment methods applicable for chemical destruction of toxic Ar-Cl derivatives are phenols as reactants much more expensive compared with sodium or potassium hydroxide, carbonate or phosphate applied in hydroxylation reactions.

Although the intramolecular C–O coupling of aryl chloride with aromatic OH group catalyzed by copper(I) salts is much easier and is applicable for the preparation of dibenzofurans (Scheme 30) [113], it could be the source of undesirable PCDD/Fs as the by-products accompanying production of polychlorinated phenols.

**Scheme 30.** Formation of dibenzofurans by Cu(I) salt of thiophene-2-carboxylic acid (CuTC) catalyzed C–O coupling [113].

### 3.3. Utilization of Ar-Xs in Cu-Mediated Biaryl Syntheses

This chapter will deal with recent developments in aryl–aryl bond formation using Ar–X as the starting material and copper and its derivatives as reagents or catalysts. Copper is the most ancient transition metal used for the synthesis of biaryls and it is still employed at present. These reactions imply the use of aromatic iodides or bromides (sometimes even chlorides) as substrates and copper metal or copper salt as the reagent. The mechanism of these reductive couplings usually involves the formation of a cuprate as the intermediate and copper halide as a byproduct (Scheme 31).

**Scheme 31.** Mechanism of Cu-catalyzed coupling of Ar–X (Ullmann reaction) [114–117].

Two molar equivalents of aryl halide are typically reacted with one equivalent of finely divided copper at high temperature (above 200 °C) to form a biaryl and a copper halide [114–116]. Considerable improvements have been made over the last century. Polar aprotic solvents such as DMF, DMSO or N-methylpyrrolidon (NMP) are the solvents that permit the use of lower temperatures and a lower proportion of copper. In addition, the use of an activated form of Cu powder, made by the reduction of copper(I) iodide with potassium, allows for the reaction to be carried out at even lower temperatures (about 85 °C) with improved yields. As the reaction is heterogeneous, it can be accelerated considerably using ultrasound halide [117]. Regarding the relationship between the structure of an aryl halide and its reactivity, electron-withdrawing groups such as nitro and carboxymethyl, especially in the ortho-position to the halogen atom, provide an activating effect. The presence of substituents that provide alternative reaction sites, such as amino, hydroxyl, and free carboxyl groups, greatly limit, however, or prevent the reaction.

Symmetrical coupling of substituted benzene rings and aromatic heterocycles using copper as the reducing and coupling agent can be performed in both inter- and intramolecular reactions.

Although steric hindrance can be a problem in Ullmann reactions, Hauser published C-C coupling of protected iodoresorcinol proceeded to afford the biphenyl in a good yield (73%) (Scheme 32) [118].

**Scheme 32.** Cu-mediated biaryl coupling of sterically hindered alkoxylated iodoresorcinol derivative [118].

Meyers et al. reported that depending on the substituent, an aryl halide can be converted to a diastereomerically enriched product under equilibrating conditions and that diastereomerically enriched biaryls are formed using the harsh reaction conditions required for the performance of a $Cu^0$ mediated biaryl synthesis [119–125].

A modern intramolecular version of Ullmann C-C bond formation under relatively mild reaction conditions was described recently. The formation of the Ar-CuBr compound was proposed as the key intermediate for the effective substitution of bromide in the Ar-Br structure. Hydroxyl-bearing amino acid **L11** (Figure 3) was identified as the efficient ligand for this C-C coupling (Scheme 33) [126].

**Scheme 33.** Example of the modern variant of intramolecular Cu-catalyzed arylation [126].

At least some water-soluble arylhalides such as halogenated aromatic sulfonic acids and their salts (bromamine acid) are capable to react with copper powder via Ullmann biaryl reaction even in aqueous solutions (Scheme 34) [127–129].

**Scheme 34.** Cu-mediated Ullmann C–C coupling of bromamine acid (dye intermediate) in aqueous solution [127–129].

Using copper(I)-thiophene-2-carboxylate (CuTC) as a promoter, Liebeskind et al. [130] reported a CuTC-mediated Ullmann-reductive coupling of substituted aromatic iodides or bromides and 2-iodoheteroaromatics at room temperature. The CuTC-mediated biaryl reductive coupling required a polar, coordinating solvent such as *N*-methylpyrrolidinone (NMP), in all probability to generate reactive Cu(I) monomers from the insoluble Cu(I) carboxylate polymer.

The authors assume that the efficiency of CuTC is probably not due to internal coordination from the sulfur atom to the metal but may be due to the inherent ability of carboxylate as a ligand to stabilize the oxidative addition product (Scheme 35). The CuTC-mediated reaction was quite general and tolerant of various functional groups. The intramolecular reductive coupling has also been successfully studied. The easy preparation of CuTC, its handling in air, and the high yields of reductively coupled products achieved under mild reaction conditions could make CuTC or other Cu(I) carboxylates the reagents of choice for many Ullmann-like reductive coupling reactions [113,131].

**Scheme 35.** Proposed mechanism of CuTC-based C–C coupling catalysis [130].

## 4. Cu-Based Hydrodehalogenation of Aromatic Compounds

Hydrodehalogenation (HDH) involves hydrogenolysis of C-X bonds, requires an external supply of reductant (usually a source of hydrogen) and is typically promoted using a metal catalyst (Scheme 36) [132].

**Scheme 36.** Scheme of common hydrodehalogenation (HDH) of Ar-X.

Although HDH is used as a reaction for the removal of halogen used as the protecting group in organic synthesis in some cases [133–137], HDH is primarily considered a developing technology for the degradation of chlorinated aquatic pollutants [132,138]. This process leads to chlorine-free hydrogenated compounds which allow for a considerable reduction in effluent toxicity. Moreover, the system can be operated under ambient conditions within a wide range of initial pollutant concentrations. The main advantage of hydrodechlorination over advanced oxidation techniques (AOPs) is the low consumption of expensive reagents. AOPs usually require pollutant mineralization for effective treatment (elimination of hazardous adsorbable organically-bound halogens AOX by-products), while the HDH process only changes the pollutant chemical structure to make them less toxic and/or more easily biodegradable.

HDH is an effective alternative procedure especially for decomposing of low concentrations of waste halogenated organic compounds under relatively mild conditions without the formation of the toxic by-products [139]. This emerging technique could have a high potential for sustainable treatment of contaminated water streams and solid wastes. Catalytic hydrodechlorination of a low concentration of halogenated aromatic compounds in contaminated liquids such as aqueous solutions or mineral oils with noble-metal and transition-metal catalysts is a particularly simple and efficient method [140]. Molecular hydrogen is often used as a hydrogen source in catalytic dechlorination. Hydrogen-transferring reactions using hydrogen donors, such as secondary alcohols and formic acid salts, have also been studied extensively. The application of noble metals catalysts is limited, however, due to the rapid deactivation of these catalysts [138,141].

The effective role of copper-based catalysis in hydrodechlorination (HDC) processes applicable for wastes contaminated with low concentrations of polychlorinated pollutants has been described [142–158].

### 4.1. Cu-Catalyzed HDC of Chlorinated Benzenes

Dechlorination of polychlorinated benzenes in the presence of copper was studied by Ghaffar and Tabata [144] at a temperature around 90 °C using isopropylalcohol and aqueous NaOH and/or Ca(OH)$_2$ in the presence of sulfur and fly ash (Scheme 37).

**Scheme 37.** HDC combined with nucleophilic aromatic substitution of chlorinated benzenes caused by the combined action of Cu/fly ash with NaOH/S/Ca(OH)$_2$ in aqueous isopropanol [144].



The observed decrease in dechlorination at a higher temperature (120–150 °C) was explained by vaporization of used solvents accompanied by a decrease in $OH^-$ and $SH^-$ nucleophiles and by the possible catalytic effect of copper in co-action of produced $Cl^-$ ions and oxygen which perform the further chlorination of produced less chlorinated benzenes at a higher temperature [68].

### 4.2. High-Temperature Cu-Mediated PCDD/Fs Dechlorination

Several research groups have demonstrated that copper-catalyzed dechlorination of PCDD/PCDF or other polychlorinated aromatic compounds proceeds on fly ash at a temperature above 250 °C with the exclusion of oxygen but without the addition of an external reductant. The observed dechlorination process was explained not only by biaryl formation via the Ullmann reaction on a copper surface, but even by the HDC reaction. In this context, the carbon matrix presented in fly ash was proposed as the reductant fundamental for low valent Cu(I) or Cu(0) capable of promoting HDC reaction at above 120 °C in the absence of oxygen [145–150].

This catalytic property of copper compounds contained in fly ash might be applied for decontamination of PCBs, PCNs and PCDD/Fs contaminated fly ash during industrial-scale operations.

### 4.3. Cu-Mediated HDC of Ar-Cls Dissolved in an Aqueous Solution

#### 4.3.1. HDC of Chlorinated Phenols

(Poly)chlorinated phenols are common contaminants of industrial water streams due to their intensive production and application as biocides and due to their relatively high solubility in water. They are in most cases produced by nucleophilic aromatic substitution of corresponding chlorobenzenes which are the next frequent pollutant of industrial wastewaters. Iron particles covered with copper are popular in remediation chemistry for the HDC treatment of water streams contaminated with halogenated compounds.

Zerovalent iron (ZVI) has been explored extensively over the last few decades for degrading and remediating a wide variety of halogenated compounds from an aqueous environment. The main drawback of the ZVI is the reported low reductive reactivity toward most of Ar–Xs.

The precious metal (usually transition metal is known as effective hydrogenation catalyst Pd, Pt, Ni, Cu, etc.) appended to ZVI, resulted in the formation of infinite galvanic cells to accelerate atomic hydrogen production [151].

Treatment of electropositive metals with more electronegative (precious) metal salts (e.g., Cu(II), Ag(I), Au(III)) can form a metal/metal galvanic cell via the cementation process (electropositive metal particles covered with a layer based on precious metal). In the metal/metal galvanic cell produced via the cementation process, the electropositive metal corrodes and transfer electrons to an electronegative metal surface where dehalogenation occurs. As one of the most abundant and low-cost metals, copper has been intensively studied for modifying zero-valent iron.

Choi et al. documented that zero-valent iron (ZVI) is completely non-reactive in the case of 2,4,6-trichlorophenol (2,4,6-TCP) in 100 mg TCP/L aq. solution treated with 100 g $Fe^0$/L. Iron metalized with Cu (Cu/Fe) causes HDC with $k_{obs} = 0.31 \times 10^{-2}\,h^{-1}$ under the same conditions. The HDC activity is explained by the acceleration of galvanic oxidation of Fe in Cu/Fe bimetal. Cu can accelerate the electrochemical reduction reaction mediated by $Fe^0$ surface due to a large potential difference between Fe ($E^0 = -0.44$ V) and Cu ($E^0 = 0.36$ V) [152].

In agreement with the observed positive effect of copper deposited on an iron surface, Su et al. described the significant catalytical effect of the $CuSO_4$ addition to the aqueous iron slurry for rapid dechlorination of hexachlorobenzene (HCB) in comparison with iron itself. The enhancement effect of the presence of Cu(II) ions on the significant increase of rate constant of HDC of HCB was attributed to the reduced Cu coating on iron particles

together with the decrease in solution pH at the start of the experiments caused by the acidic CuSO$_4$ addition [153].

In contrast, the high removal efficiency of pentachlorophenol from the aqueous solution after 2–3 days of Cu/Fe bimetal action was explained by chemisorption of chlorophenolates on the metal oxide surface of Cu/Fe by Kim and Carraway rather than the HDC reaction [154].

It was also observed by Duan et al. [155] that copper works as the hydrodechlorination agent for 4-chlorophenol (4-CP) more rapidly (with $k_{obs}$ = 1.37 × 10$^{-1}$ h$^{-1}$) than bimetals Cu/Ni or Cu/Fe under conditions when 0.4 mM 4-CP was treated by 100 g/L of the respective metallic reductant. In addition, cyklohexanone and soluble Cu(II) salt are produced by HDC of 4-CP using Cu. HDC based on the action of Cu/Fe bimetal produces phenol na Fe(II) salt much more slowly (with $k_{obs}$ = 8.61 × 10$^{-3}$ h$^{-1}$). The authors deduced different mechanisms of Cu and Cu/Fe actions based on these observations (Scheme 38).

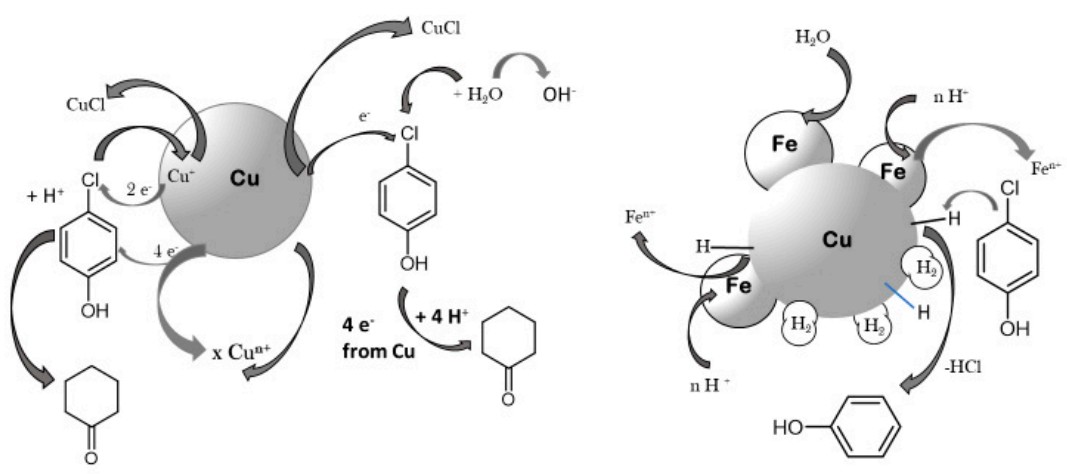

**Scheme 38.** Proposed two HDC mechanisms of 4-CP using a metallic copper and bimetallic Cu/Fe system [155].

Apart from the above-mentioned co-action of Cu in HDC, the catalytic HDC action of copper was observed after the addition of sodium borohydride as the reductant. Rapid HDC of several chloroaromatics dissolved in aqueous solution (34 mg Ar–Cl/L) was observed using nanocopper (2.5 g Cu/L) and NaBH$_4$ (1 g/L) (Scheme 39) [156].

**Proposed HDC mechanism of PhCl [156]:**

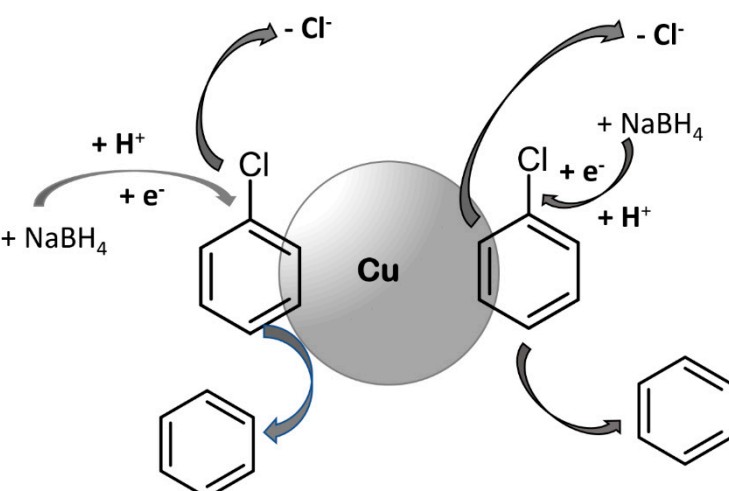

**Scheme 39.** Cu-catalyzed HDC of chlorobenzene using NaBH$_4$ as the reduction agent [156].

Similar to the above mentioned HDC mediated by Cu using NaBH$_4$ reductant, the HDC of DDT (50 ppm in H$_2$O) and 2,4,6-TCP (50 ppm in H$_2$O) was achieved using a mixture of Devarda's alloy (50% Cu + 45% Al + 5% Zn in quantity 0.5 g/L) and sodium borohydride alloy (1 g NaBH$_4$/L) at 80–100 °C [157].

Partial HDC of polychlorinated benzenes and regioselective HDC of some chlorinated phenols and chlorinated anilines using Devarda's Al–Cu–Zn alloy in an alkaline aqueous solution at room temperature was studied by Weidlich et al. [158]. The HDC activity was explained by the galvanic oxidation of Al from Devarda´s Al–Cu–Zn alloy in an aqueous 1 wt.% NaOH solution.

### 4.3.2. Cu-Catalyzed Hydrodebromination (HDB) of Polybrominated Phenols

Flame retardants are common organic specialty chemicals widely used to improve the flame resistance in circuit boards and plastics. Broadly used flame retardants based on brominated phenols (brominated flame retardants, BFRs) have been of concern recently for our environment because of their endocrine disruptive, immune toxicity and risk of toxic brominated byproducts formation. They are ubiquitously present in the electrical wastes (e-wastes) and persist at low concentrations [159–161].

Pyrolysis and combustion are often applied to dismantle e-wastes. The incineration of BFRs-containing e-wastes usually produces, however, polybrominated dibenzo-p-dioxins (PBDDs) and dibenzofurans (PBDFs), which are highly toxic in the environment [162–164].

As a potential remediation technology, hydrodebromination (HDB) of persistent pollutant tetrabromobisphenol A (TBBPA) from both aqua and soil by zero-valent iron particles (nZVI) covered with transition metals was intensively investigated (Scheme 40) [165,166].

**Scheme 40.** Hydrodebromination (HDB) of TBBPA in contaminated aqueous solution using an excess of bimetallic particles.

The study of Li et al. demonstrates that deposition of Cu on nZVI surface drastically improved the debromination of tetrabromobisphenol A (TBBPA) [167]. This technique utilizes Fe(III) and Cu(II) salts as metal sources and NaBH4 as a powerful reductant for the initial preparation of nZVI (Equation (4)), which is subsequently covered by reductive deposition on Fe(0) after the addition of Cu(II) salt (Equation (5)).

$$4\ Fe^{3+} + 3\ BH_4^- + 9\ H_2O \rightarrow 4\ Fe^0\ 3\ H_2BO_3^- + 12\ H^+ + 6\ H_2 \tag{4}$$

$$Cu^{2+} + Fe^0 \rightarrow Cu^0 + Fe^{2+} \tag{5}$$

Cu-nZVI with a low dosage could rapidly and completely debrominate TBBPA into BPA. This is attributed to the above-mentioned excellent galvanic effect in the presence of Cu. It was reported that the proper amount of Cu islets coating on the nZVI surface and a weakly acidic condition favored the fast rate (ca. $3.41 \times 10^{-2}$ min$^{-1}$) of debromination and the great extent of TBBPA degradation. The oxidation of Fe$^0$ and the elevation of pH promoted the co-precipitation of Fe(II) and Fe(III) and thus the formation of amorphous magnetite on the Cu–nZVI surface. This study suggests that Cu–nZVI can be effectively used to remove TBBPA from both water and soil, reducing the toxicological effects of TBBPA in aquatic and edaphic environments [167].

Even sole copper nanoparticles are an effective debromination agent for reduction of TBBPA to BPA even in co-action with Fe(II) salts occurring in green rust. Fe(II) works as a reductant for the regeneration of active Cu$^0$ species [168]. This observation corresponds well with the above-mentioned reductive potential of metallic copper for HDC of 4-CP [155].

Cu-based debromination of TBBPA was even observed during pyrolysis of electronic waste. Cu has a significant catalytic effect on the pyrolysis process and products. Cu can promote the conversion of the pyrolytically produced organic bromides such as bromo-methane, bromoethane, 2-bromophenol, and 2,4-dibromophenol to Br$_2$ and HBr. The conversion mechanism of bromide species is simplified as the preferential adsorption of organic bromides followed by the coordination of bromine atoms and Cu. The electron transfer results in the cleavage of the C–Br bond and the generation of Br$_2$ or HBr. In contrast, Cu can act as a catalyst to promote the debromination reaction through the Ullmann crossing-coupling reaction. Cu could therefore promote waste printed circuit boards pyrolysis and make the pyrolysis products easier to dispose of to some extent [169].

We observed that 2,4,6-tribromophenol (2,4,6-TBP) even in a quite high concentration dissolved in alkaline aqueous solution (25 mM of 2,4,6-TBP in 1.5 wt.% aq. NaOH) underlies rapidly to HDB using Devarda´s Al-Cu-Zn alloy (18 g/L) after 2 h at room temperature and phenol is produced quantitatively (Scheme 41). This reaction is slower than HDB promoted by Raney Al-Ni alloy but the efficiency is the same. In addition, copper is not as toxic an element in comparison with nickel. The filtrate obtained after neutralization which contains phenol is completely biodegradable by bacteria *P. fluorescence* or *R. erythropolis*, as we verified [170,171].

**Scheme 41.** Cu-catalyzed HDB of 2,4,6-TBP dissolved in aq. NaOH mediated by Devardas Al–Cu–Zn alloy [170,171].

### 4.3.3. Cu-Catalyzed HDB of Polybrominated Diphenylethers

Polybrominated diphenylethers (PBDEs) are well-known BFRs and are widely used in circuit boards, furniture, plastics, textiles, and other commercial products designed for fire protection [172]. PBDEs belong to a class of persistent organic pollutants (POPs), with potential toxicity to the liver, the reproductive system and the development of mammals. PBDEs have been found in both, natural and artificial environments (such as sediments from the rivers, lakes, and oceans) [173]. In addition, during pyrolysis of polybrominated diphenylethers (PBDEs), which are used as flame retardants in synthetic polymer blends, polybrominated dibenzo-p-dioxins PBDDs and polybrominated dibenzofurans PBDFs are formed similarly to TBBPA [174].

There are a number of studies on the degradation of PBDEs by chemical reduction and many in-depth reaction mechanisms have been revealed. Compared to a chemical reduction, photolysis for PBDE degradation involves the dissipation of a high amount of energy and is costly [175].

In comparison, a chemical reduction is considered to be the most feasible solution to the restoration of the PBDEs-contaminated area. Debromination of PBDEs by (nZVI) or iron-based bimetals has become the primary mode of chemical reduction (Scheme 42) [176]. According to the significantly different debromination activity of Fe/Cu bimetal in electrically conductive water and non-conductive ethanol, the authors suggested that the effect of galvanic couple formed by covering of $Fe^0$ surface with $Cu^0$ (electron-transfer) plays the dominant role in 2,2′,4,4′-tetrabromodiphenyl ether (BDE-47) debromination mechanism. In addition, Fe/Cu reduces BDE-47 finally to diphenylether (DPE) and no toxic dibenzofuran was produced during this process [177].

**Scheme 42.** HDB of 2,2′,4,4′-tetrabromodiphenylether (BDE-47) using bimetallic Cu/Fe particles [176–178].

The comparison of HDB kinetics determined the high debromination activity of metallic copper both in the form of Fe/Cu bimetal and copper in co-action of reductants such as $H_2$ or $NaBH_4$. In addition, the determined debromination rate using Fe/Cu is lower in comparison with Fe/Pd or Fe/Ag but higher than the debromination rate of Fe/Ni, Fe/Au or Fe/Pt bimetals or Fe (ZVI) [178].

Although metallic copper in few cases promotes HDH reaction better in comparison with Cu/ZVI, in most cases effective application of metallic copper in co-action of a reductant such as iron, aluminium or $NaBH_4$ is meanwhile the method of choice for effective HDH of (poly)halogenated phenols in contaminated aqueous solutions. Especially combination of copper disposing with high specific surface area and powerful but in aqueous solution also enough stable reductant as is $NaBH_4$ potentially offers options for new findings in HDH processes.

## 5. Conclusions

This review summarizes recent developments in the field of copper-catalyzed C$_{arom}$–H halogenation and subsequent C$_{arom}$–X transformation reactions, mainly focusing on C$_{arom}$–X substitution by O-nucleophiles, C-C couplings and hydrodehalogenation which are most significant in the chemical treatment of toxic polyhalogenated aromates such as polybrominated diphenyl ethers or polybrominated phenol, polychlorinated biphenyls, polychlorinated dibenzo-p-dioxins and dibenzofurans.

Copper-catalyzed reactions have gained significant attention because of its cost effectiveness, low toxicity of Cu and perception of copper as an environment-friendly catalyst widely used in organic chemistry. Moreover, it is absolutely clear from the reported work that most of the methods belong to cheaply-available catalysts such as metallic copper, copper halides, oxides and copper acetate as the catalytic systems. The Cu-catalyzed direct C$_{arom}$–H halogenation reaction has emerged as a powerful strategy for the synthesis of a wide variety of halogenated aromatic compounds. With the utilization of an appropriate directing group, Cu-catalyzed halogenation enables selective preparation of sterically hindered, but rarely accessible ortho-halogeno-substituted aromatic compounds.

One of the most important Cu(II) catalyzed oxidation reactions influencing not only industrial production of Ar-Xs but even the undesirable formation of persistent polyhalogenated by-products is the air oxidation of HX (oxyhalogenation). This reaction is responsible for the formation of stable and toxic polyhalogenated aromatic compounds such as PCBs and PCDD/Fs during industrial thermal treatment processes in the presence of oxygen (metal ores sintering, de novo synthesis during waste combustion, etc.). Since the formation of undesirable PCBs, PCDD/Fs or PCNs is based on the utilization of the copper

surface, several inhibitors of de novo synthesis have been proven e.g., some amines or sulfur-containing compounds which act as blockers and deactivators of the catalytically active copper surface [179–185]. The prospective application of these inhibitors could be potentially the crucial practice integrated into the key technological processes (waste incineration or metallurgy), which are at fault for PCDD/Fs or PCBs production, for minimization of PCBs, PCDD/Fs or PCNs genesis.

In contrast, the catalytic activity of Cu and its salts at an elevated temperature permits straightforward cleavage of C$_{arom}$–X bonds producing corresponding non-halogenated aromatic products not only in a synthesis of organic fine chemicals but even in very effective dehalogenation processes applicable in the area of halogenated waste treatment and remediation processes. The discovery of the rate acceleration effect of organic ligands in promoting cross-coupling reactions has expanded the scope of copper catalysis to new areas, promising novel reactivities and selectivities. Aryl halides are the most explored substrates in transition metal-catalyzed cross-coupling reactions. Copper-based catalysts demonstrated good activity and selectivity for all types of C$_{arom}$–X substitution reactions of aryl iodides. Overall, aryl bromides and heteroaryl bromides manifested moderate reactivities, while aryl chlorides have generally been considered inert to copper-based catalysts. The discovery of new ligands and the application of more reactive Cu-based nanocatalysts have enabled, however, cross-couplings to be achieved in some cases with aryl chlorides. In the future, the ultimate goal should be the design and further development of convenient copper-based catalysts for the activation of inert aryl chloride substrates. In this area, new methodologies employing tandem catalysis to transform unreactive aryl chloride into a more reactive aryl iodide as an intermediate that subsequently undergoes a variety of Cu-based coupling reactions could be a promising area not only in terms of the broader exploitation of copper-catalyzed reactions to large-scale synthesis of organic fine chemicals [186,187], but even for the development of simpler destruction techniques available for the treatment of PCBs, PCNs or PCDD/Fs.

Moreover, metallic copper in the co-action of appropriate reductants, especially NaBH$_4$, Fe or Al, makes possible facile destruction of polybrominated and polychlorinated aromatic pollutants e.g., polyhalogenated phenols and polybrominated diphenyl ethers in contaminated soils and wastewater streams. A huge excess of reagents is still required, however, especially for effective hydrodechlorination of chlorinated aromatic pollutants. These research directions can therefore be a rich area for further future investigation.

**Funding:** This research received no external funding.

**Acknowledgments:** This research was funded by the Faculty of Chemical Technology, University of Pardubice, within support of an excellent research team Chemical Technology Group.

**Conflicts of Interest:** The author declares no conflict of interest.

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
