# Peer review of "The Influence of Copper on Halogenation/Dehalogenation Reactions of Aromatic Compounds and Its Role in the Destruction of Polyhalogenated Aromatic Contaminants"

_catalysts, doi:10.3390/catal11030378_

Round 1

Reviewer 1 Report

Overall, this Review by Weidlich is a good resource for those interested in copper-catalyzed halogenation/dehalogenation reactions in the context of both environmental and synthetic chemistry. My only general suggestion is that other forms of C–X to C–pseudohalide reactions under copper catalysis are discussed to some extent such as cyanation. I recommend that a read through for grammatical corrections is performed prior to publication.

Author Response

The author is grateful to the reviewer for his time, valuable comments and suggestions which helped to improve this manuscript.

Author’s reply:

I have added additional information dealing with the possible effect of other nucleophiles in reaction according to Schemes 6-7 including TMSCN (please, see below).

In general, this review deals with methods applicable for more possible simple chemical treatment of polyhalogenated aromatic compounds transforming them to less harmful products. I argue that substitution of Ar-Xs with pseudohalides (including cyanides) does not yield sufficient advantages compared with O-nucleophiles in this area (I used information herein published for example in Li et al: Org Proc. Res. Dev. 2017, 21, 1889-1924). This is the main reason why the reaction of pseudohalides was not discussed in more detail in this review.

This manuscript was edited and corrected by a native speaker. I apologize for residual grammar errors and hope they have now been repaired.

Reviewer 2 Report

In this manuscript, the author reviewed the halogenation/dehalogenation reaction using copper. It is an important research topic and the author corrected many related papers, although, this review is just a list of results and lack of consideration for new insight. The reviewer approves the publication of this paper after adding some discussion summarizing each reaction.

Minor points;

  1. Line 122: There is no reaction about Ewg substituted phenol derivatives in reference [36]. So the author should rewrite this context.
  2. Scheme 4&5: Edg should be changed such as X. It (Edg-H) is confusing.
  3. Scheme 4: The stoichiometry of Br- does not match.
  4. Scheme 4, caption: reference no. [37] should be added.
  5. Scheme 12: The author should explain the mechanism of this one-pot acylation halogenation reaction.
  6. Line 327: Scheme 23 should be changed to Scheme 21

Author Response

The author is grateful to the reviewer for his time, valuable comments and suggestions which helped to improve this manuscript.

Thank you very much for your help. 

I hope that the other described reactions are discussed sufficiently for possible subsequent research considerations. The crucial currently achieved milestones, in each of the discussed directions of research, are introduced in the corresponding (sub)chapters, the conclusions and in the summary of the potential considerations in the research areas is mentioned in the Conclusion chapter.